# Prenatal development of neonatal vocalizations

**Darshana Z Narayanan[1,2], Daniel Y Takahashi[1,2]\*, Lauren M Kelly[1,2], Sabina I Hlavaty[3], Junzhou Huang[4], Asif A Ghazanfar[1,2]\***

[1]Princeton Neuroscience Institute, Princeton University, Princeton, United States; [2]Department of Psychology , Princeton University, Princeton, United States; [3]Department of Ecology and Evolutionary Biology, Princeton University, Princeton, United States; [4]Department of Computer Science and Engineering, The University of Texas at Arlington, Arlington, United States

**Abstract** Human and non-human primates produce rhythmical sounds as soon as they are born. These early vocalizations are important for soliciting the attention of caregivers. How they develop remains a mystery. The orofacial movements necessary for producing these vocalizations have distinct spatiotemporal signatures. Therefore, their development could potentially be tracked over the course of prenatal life. We densely and longitudinally sampled fetal head and orofacial movements in marmoset monkeys using ultrasound imaging. We show that orofacial movements necessary for producing rhythmical vocalizations differentiate from a larger movement pattern that includes the entire head. We also show that signature features of marmoset infant contact calls emerge prenatally as a distinct pattern of orofacial movements. Our results establish that aspects of the sensorimotor development necessary for vocalizing occur prenatally, even before the production of sound.

**\*For correspondence:**
takahashiyd@gmail.com (DYT);
asifg@princeton.edu (AAG)

**Competing interest:** The authors declare that no competing interests exist.

## Editor's evaluation

This paper will be of great interest to the field of developmental neuroscience and social communication. The authors were able to identify sensorimotor vocal precursors in fetal marmoset monkeys by using ultrasound imaging to detect rhythmic orofacial movements related to vocalizations. These findings provide new insights into the prenatal development of vocal behavior in primates. The data acquired by a highly quantitative approach support the major claims of the paper.

## Introduction

Neonatal primates produce vocalizations (e.g. cries and coos) as a rhythmical series of sounds (*Hopkins and von Wulfften Palthe, 1987*; *Wolff, 1969*). These vocalizations are often the substrate for the rhythmic vocalizations produced by adult humans (*Chandrasekaran et al., 2009*) and other primates (*Risueno-Segovia and Hage, 2020*; *Lameira et al., 2015*; see *Ghazanfar and Takahashi, 2014* and *Bergman et al., 2019* for reviews). Their production requires coordination among all the elements of the vocal apparatus, i.e., between laryngeal tension, respiration, and orofacial movements. We know that postnatal vocal production is a complex motor act that changes over time and is influenced by caregiver feedback (*Goldstein and Schwade, 2008*; *Warlaumont et al., 2014*) and the ambient environment (*Mampe et al., 2009*). But how do those very first neonatal vocalizations develop?

During human pregnancies, movements consistent with crying have been observed during ultrasound assessments of fetuses in their last trimester (*Gingras et al., 2005*) but their prenatal developmental trajectory and differentiation from other movements have not been measured. We reasoned

**eLife digest** Much like human babies, newborn monkeys cry and coo to get their caregiver's attention. They all produce these sounds in the same way. They push air from the lungs to vibrate the vocal cords, and adjust the movement of their jaws, lips, tongue and other muscles to create different kinds of sounds.

Ultrasounds show that human fetuses begin making crying-like mouth movements during the last trimester of pregnancy. Yet the prenatal development of this crucial skill remains unclear, as most studies of early primate vocalization take place after birth.

To explore this question, Narayanan et al. focused on a small species of monkeys known as marmosets. Regular ultrasounds were performed on four pregnant marmosets, starting on the first day the fetuses' faces became visible and ending the day before delivery. The developing marmosets acquired the ability to independently move their mouth from their head over time, a skill crucial for feeding and vocalizing. By the end of pregnancy, a subset of fetal mouth movements were nearly identical to those produced when baby marmosets call for their caregivers after birth.

Human ultrasound studies are needed to confirm whether vocal development follows a similar trajectory in our species. This is likely given the developmental similarities between both species. If so, work in marmosets could be helpful to understand how conditions such as cerebral palsy interfere with this process, and to potentially develop early interventions.

that a nonhuman primate model of this behavior would be a way to gain insights. Marmoset monkeys are a good candidate model for human infant vocal behavior. The early postnatal vocalizations of infant marmosets undergo a very similar developmental trajectory as infant human vocalizations: (1) they transition from noisy cries to tonal vocalizations that are adult-like (*Takahashi et al., 2015*); (2) these early vocalizations are produced in bouts that have a rhythmic pattern that changes over time (*Zhang and Ghazanfar, 2016*); and (3) vocal development is influenced by parental care and feedback (*Takahashi et al., 2017*; *Gultekin and Hage, 2017*; *Gultekin and Hage, 2018*). These across-species similarities in postnatal vocal development are also occurring at the same life history stage (though marmosets develop 12 times faster than humans [*de Castro Leão et al., 2009*]), supporting the idea that this species may also be a good model for prenatal human behaviors.

We tracked the fetal orofacial movements of marmoset monkeys that potentially represent those movements necessary for their neonatal contact-calling. These contact calls consist of both immature and mature-sounding versions (but mainly the former early in postnatal life) (*Takahashi et al., 2015*) and whose orofacial movements are indistinguishable from each other. Importantly, neonatal marmoset contact vocalizations consist of orofacial movements that are spatiotemporally distinct from any other vocalization in the marmoset's vocal repertoire. They therefore provided a template: we could compare prenatal mouth movements with those produced during neonatal contact calling. We quantified and characterized the prenatal trajectory of these movements to address the following questions: if and how do mouth movements differentiate from more global bodily movements, and if and how do mouth movements differentiate prenatally into those used for neonatal vocalizations?

## Results

We performed non-invasive ultrasound imaging on awake, pregnant marmoset monkeys (n=4 pregnancies across two marmoset monkeys). Marmosets typically produce dizygotic twins at birth but not always (*Harris et al., 2014*). In our sample, the four pregnancies consisted of one singleton, two sets of twins and one set of quadruplets. Since individual fetuses cannot be routinely identified via ultrasound, each pregnancy with more than one fetus was treated as a composite of a single subject. To track developmental trajectories, we densely sampled fetal orofacial movements from the day the face was clearly discernible to the day before birth (~embryonic day [E]95–146; 2–3 times per week with imaging sessions lasting between 15 and 45 min; 14–17 sessions per pregnancy; a total of 64 sessions). We then compared these movements to signature features of infant vocalizations in the first week after birth (postnatal day [PD] 1–7; n=7). We were thus able to observe the developmental trajectory of orofacial movements necessary for vocal production from the fetal to infant stage.

We had two basic hypotheses. The first was that fetal orofacial movements would initially be linked to movements of other body parts. Ultrasound studies of human fetuses support this hypothesis, showing that isolated jaw movements appear late in development, weeks after the onset of spontaneous general movements involving many body parts (*Fagard et al., 2018*; *Kurjak et al., 2004*). Our second hypothesis was that signature patterns of articulatory movements related to postnatal marmoset contact calls would emerge prenatally. Investigations of the prenatal movements of rats, and prehatching movements of birds, show that seemingly unorganized action patterns are followed by organized action patterns that are reminiscent of postnatal or post-hatching movements. (*Bekoff and Lau, 1980*; *Brumley and Robinson, 2010*; *Smotherman and Robinson, 1988*; *Bradley, 1999*; *Hamburger, 1963*; *Kuo, 1932*; *Provine, 1980*).

In each imaging session, we specifically targeted the face as our region of interest, which allowed us to track both orofacial movements and head movements. From these videos, we did a frame-by-frame analysis (*Figure 1A*). The developmental change was readily noticeable, and ostensibly in keeping with our expectation that orofacial and head movements move together more often in the younger compared to the older fetus (*Figure 1B–C*; *Videos 1 and 2*).

One of the hallmarks of development is increasing 'order' in behavior through self-organization. Since the relationship between fetal head and orofacial movements in marmosets changed over time, we tested if, for these two movement behaviors, there is concomitant increase of structure in their action patterns, by mapping the sequence of five different states: independent orofacial movements (State 1); independent head movements (State 2); orofacial movements followed by overlapping head movements (State 3); head movements followed by overlapping orofacial movements (State 4); and synchronous orofacial and head movements (State 5). The fetal behavioral states are seen to be more numerous and variable in the early weeks of gestation, when compared to late gestation. *Figure 2A* shows the proportion of states through gestation for the population data; *Figure 2B* shows the state diagrams for exemplar sessions selected from different gestational periods of one pregnancy.

We quantified these state changes across gestational days in two ways. First, Shannon entropy was used to measure behavioral variability within each session (*Figure 2C*). For determining the developmental trajectory, the best polynomial-fit order using Akaike's information criterion (AIC) was found to be three. Behavioral variability decreased through gestation and then remained steady at ~0.8 bits (the maximum entropy for a behavior with 5 possible states is 2.32 bits). The decrease in entropy ($p < 0.001$, test of nullity of the relation between gestational day and entropy) indicates that as the fetus gets older, there is increasing movement structure.

Second, we performed a Kullback-Leibler divergence test to quantify the behavioral change through fetal life (*Figure 2D*). The average state distribution of E93-99 (n=6) was compared with the state distributions of all the imaging days. The best polynomial-fit order for the divergence estimates was one. The resulting linear fit with a positive slope ($p < 0.001$) indicates that with increasing gestational age, fetal behavior—with respect to orofacial and head movements—becomes increasingly different from the first imaging day of orofacial movement. Thus, what we see is that in the young fetus, orofacial and head body parts move in a number of different ways—independently, and also often together in various combinations (orofacial-head, head-orofacial, or synchronous onset), but when close to birth, orofacial and head body parts are, with rare exception, moved separately. This suggests that these two motor regions are decoupled in the late-stage fetus, which allows the newly born infant to use head and orofacial movements adaptively; in marmoset infants, this would putatively be for the different purposes of orienting, feeding, and vocalizing.

To further quantify the developmental change, we examined the profiles of orofacial movements (n=1977) and head movements (n=1216) independently. We calculated the rates of orofacial and head movements (number of movements per hour) in each imaging session. The best polynomial-fit order was two for the orofacial movements, and one for head movements. The occurrence of orofacial movements showed an inverse U-shaped profile; steadily increasing from ~E93-105, hitting a peak from ~E106-131 and declining from ~E119-131 (*Figure 2E*, in red). The head movements, however, showed a linear trend. For both orofacial and head movements, the same patterns were observed in the individual pregnancies (*Figure 2F*; clockwise from top left: singleton, quadruplets, twins, and twins).

To study the putative 'linked' relationship between these two motor patterns, we calculated the percentage of temporal overlap between them (total instances of overlap divided by the total number

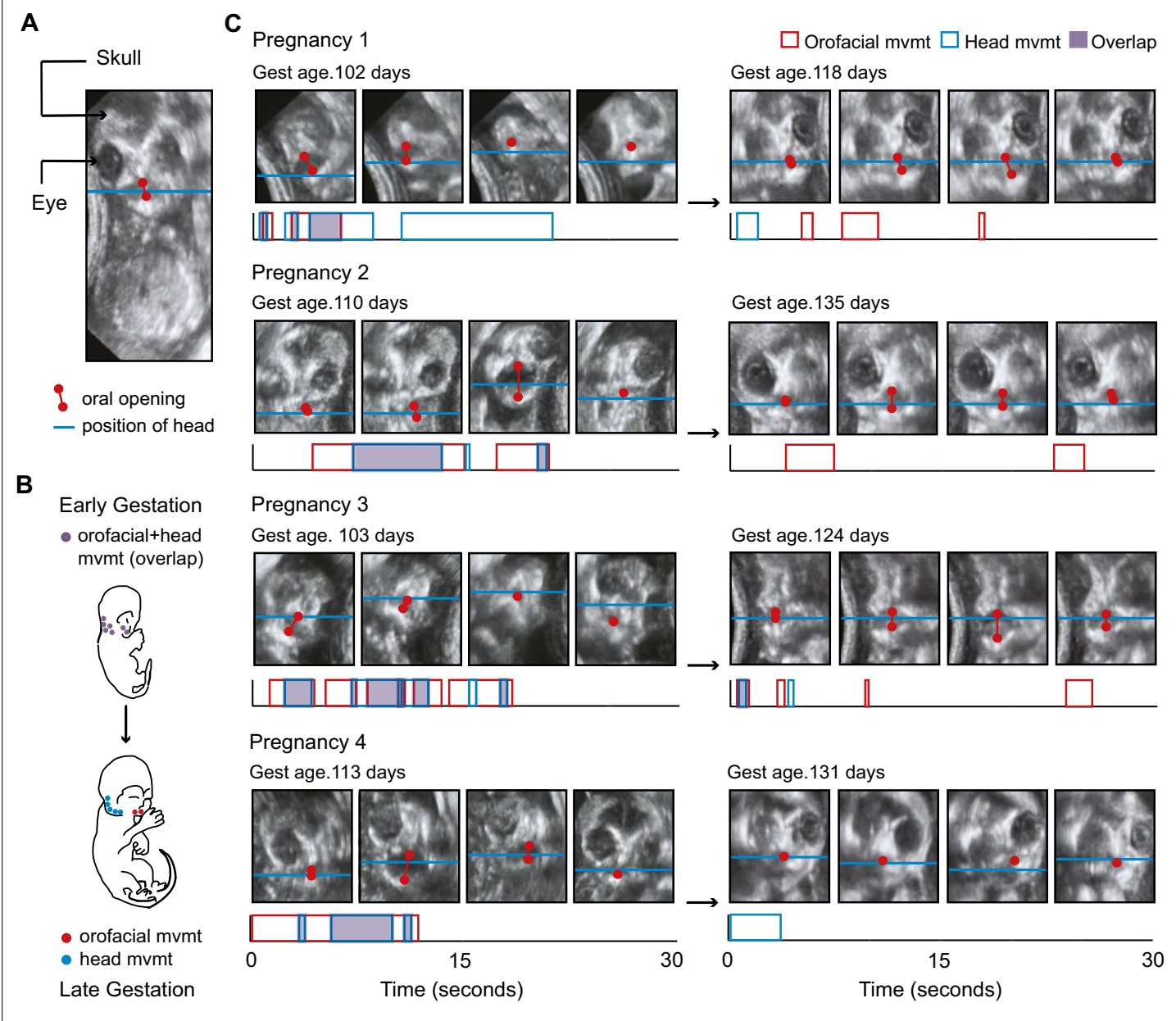

**Figure 1.** Fetal orofacial movements undergo striking changes through gestation, observable using ultrasound imaging. (**A**) Still frame from fetal ultrasound. Red is used to mark the upper and lower jaws and indicate oral opening. The blue line marks the position of the head, tracked using the jaw joint. (**B**) Illustration of the expected developmental change in orofacial and head movements. The top image depicts a young fetus with orofacial and head regions often moving together. The bottom image illustrates a late-stage fetus, with orofacial and head regions moving in isolation. (**C**) Developmental change in orofacial and head movements for each pregnancy. Top: still frames from representative ultrasound clips. Gestational day is indicated in the upper left of each panel. Below: time stamps of the orofacial and head movements in the clip. Durations of orofacial (red) and head (blue) movements, and regions of overlap (purple) are indicated. The amplitude of movement has no bearing on our analysis. This developmental change can be observed by comparing *Videos 1 and 2*.

of movements). A first order polynomial curve was found to be the best fit for determining the overlap profile. We found that the overlap of orofacial and head movements decreases over the course of fetal development (*Figure 2G* in purple, p<0.001). One possibility is that, since movements decrease in general toward the end of gestation, the decrease in overlapping movements simply becomes less probable. To account for this, we did a permutation test to be certain the decline in overlapping movements was independent of general movement decline. Orofacial and head movements within each session were independently shuffled, keeping the duration and latency distributions intact. Percent

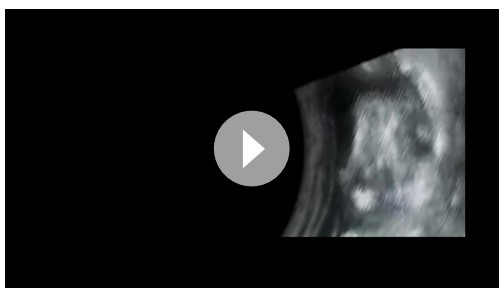

**Video 1.** Young fetus with orofacial and head regions moving together.
https://elifesciences.org/articles/78485/figures#video1

overlap was calculated for the permuted data, and a first order polynomial curve was fitted. The permutation test generated a flat profile (p=0.057 for the significance of the mean regression line) compared to the overlap decline seen in the actual data, confirming that the decline in overlap is not simply due to the decline in overall movement quantity (*Figure 2G*). Another possibility is that movement duration decreases throughout gestation, which would, therefore, decrease the probability of overlapping movements. This was not the case: the durations of mouth movements increased with time (p=0.024) and the durations of head movements remained steady throughout gestation (p=0.093). Thus, consistent with our prediction that mouth movements would be linked to other body parts, orofacial and head movements are coupled early in gestation but become increasingly independent. This held true for each of the four pregnancies (*Figure 2H*; clockwise from top left: singleton, quadruplets, twins, and twins).

Regarding vocalizations, we next focused only on the patterns of fetal orofacial movements to test the hypothesis that they gradually exhibited the temporal organization of the infant marmoset monkey's rhythmic contact calling (*Zhang and Ghazanfar, 2016*). To do this, we quantified the temporal pattern of contact calling by neonatal marmosets (within ~24 hr of birth); these were the same individuals that were imaged as fetuses (n=7; one singleton, two sets of twins, and two surviving individuals from the quadruplets). Contact calling was induced by briefly separating infants from caregivers; we acquired ~5 min of audio recordings and ~5 min of both video and audio. In both media, infant contact calls are easily identifiable by their long duration and many syllables (*Takahashi et al., 2015*; *Zhang and Ghazanfar, 2016*; *Figure 3A*). We compared the recordings of infants producing contact calls to imaging data from fetuses producing orofacial movements (*Figure 3B* shows stills from the videos of an infant at P1 and a fetus at E118). The upper and lower jaws were tracked frame-by-frame to generate temporal profiles of the movements. We found that a subset of the fetal orofacial movements was a match to the infant contact call in terms of overall temporal profile, duration, and syllable number. (*Figure 3C–F*; *Video 3*).

The cyclic nature of the infant contact call is fully captured in its temporal profile. We therefore matched the temporal profiles of fetal orofacial movements to those of infant contact calls. To do this, we took the video recordings (n=25) of neonatal marmoset infants producing contact calls and did a frame-by-frame tracking of the upper and lower jaw positions to generate contact call profiles. Multiple contact call profiles of the same syllable number were averaged and smoothed to create a set of contact call 'templates' (n=7; 1–7 syllables). To get the temporal profiles of fetal orofacial movements, we used a custom-made computer vision program that tracked each movement. Only temporal profiles that matched the manually calculated syllable number features of the movement were included in the analysis (n=414). We smoothed these profiles using a Savitzky-Golay smoothing filter (polynomial degree = 3; sliding window size = 9). We then used dynamic time warping (DTW) analysis (*Keogh and Ratanamahatana, 2005*) to measure the similarity between the temporal profiles of the fetal orofacial movements and the infant contact call templates. Every fetal orofacial movement profile was compared with every contact call template, and the one with the smallest DTW cost (i.e. the closest similarity) was chosen. The median DTW cost was calculated for each session (n=57). Our prediction was that the DTW cost would decrease through gestation, indicating that the temporal profiles of the fetal orofacial movements are getting closer to those of infant contact calls (*Figure 4A*). We found this

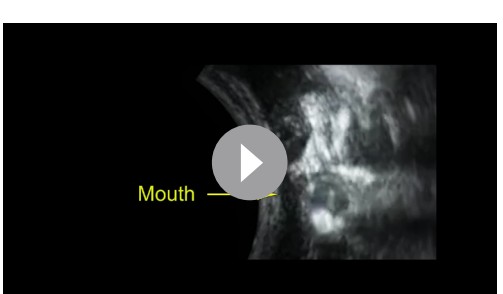

Mouth

**Video 2.** Late-stage fetus with orofacial region moving in isolation.
https://elifesciences.org/articles/78485/figures#video2

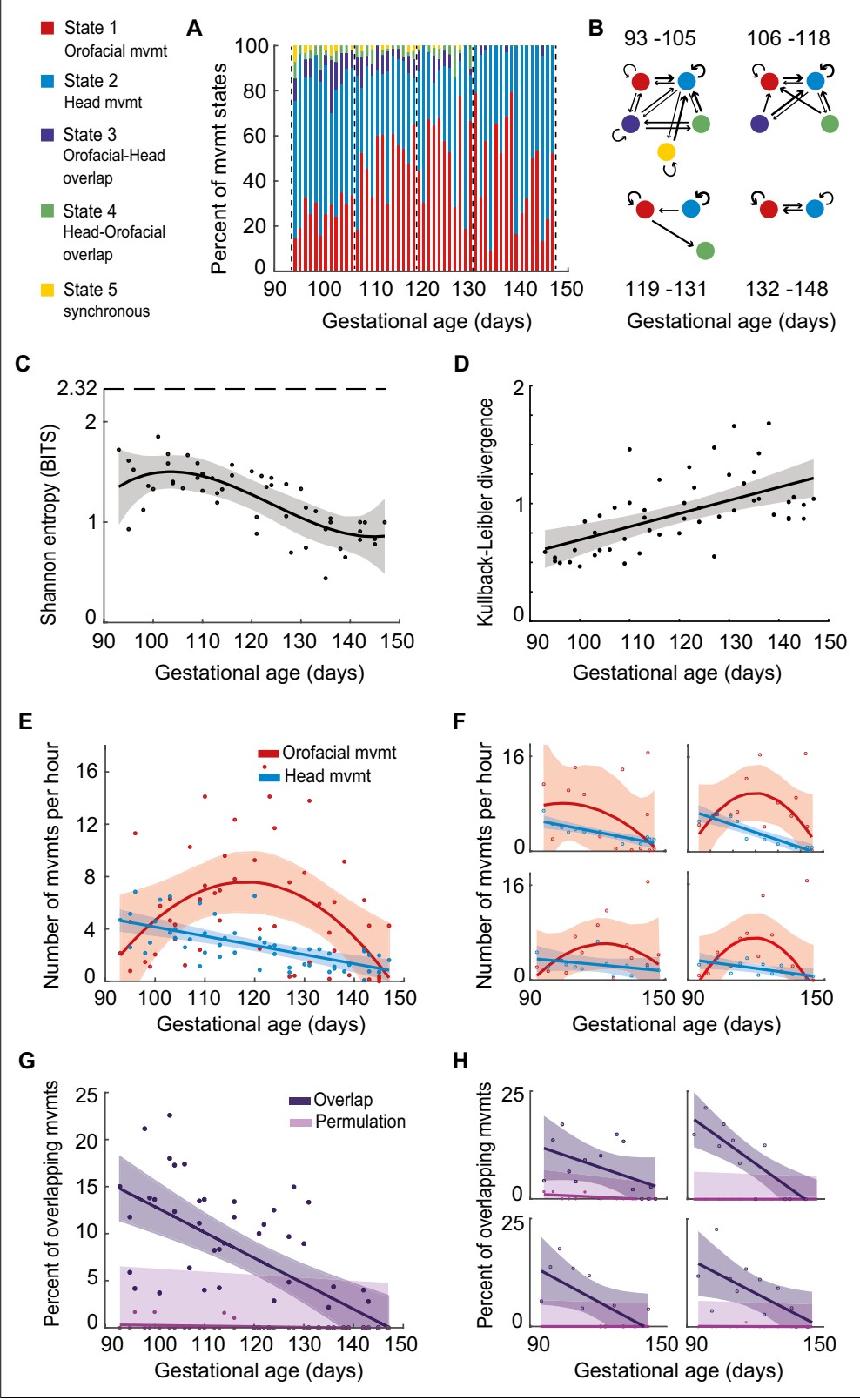

**Figure 2.** Fetal orofacial movements differentiate from a larger movement pattern that includes the head, leading to an increase in 'order' in the motor behavior of the fetus. (**A**) Proportion of behavioral states per session. A total of 5 states are represented in red (isolated orofacial movements), blue (isolated head movements), purple (overlap: orofacial followed by head), green (overlap: head followed by orofacial) and yellow (synchronous

*Figure 2 continued*

movements). (**B**) Transition diagrams visualizing the behavioral states in four sessions chosen from different stages of gestation of a single pregnancy. The widths of the arrows indicate the transition frequencies between states. (**C**) Developmental change in the behavioral variability per session. Points represent entropy measures for single sessions, and the curve represents optimal polynomial fit based on Akaike's information criterion (AIC). 2.32 bits is the maximum entropy for a behavior with 5 possible states. (n=64 sessions; p<0.001 in the test of nullity of the relation between gestational day and entropy) (**D**) Kullback-Leibler (KL) divergence of behavioral state distributions. Points represent the relative entropy measures for every session compared to the first testing session, and the curve represents optimal polynomial fit. (n=64 sessions; p<0.001 for linear fit with a positive slope) (**E**) Developmental change in the rates of orofacial and head movements. Points represent the rate of orofacial (red) and head (blue) movements in each session (n=1977 for orofacial movements and n=1216 for head movements, in 64 sessions), the curves represent the optimal polynomial fits. (**F**) Same as (**E**), for each pregnancy. (**G**) Developmental change in overlap of orofacial and head movements. Points represent the percentage of overlap in each session (n=64 sessions), curves represent optimal polynomial fits for overlap and permutation test, shaded regions denote the 95% confidence interval for the fits. (p<0.001 for linear fit with negative slope, p=0.057 for the mean regression line in the permutation test) (**H**) Same as (**G**), for each pregnancy.

to be the case (*Figure 4B*). A multiple linear regression analysis controlling for the different pregnancies shows that gestational age predicts the prevalence of 'contact call-like' temporal profiles ($\beta\pm SE$ = $-0.005\pm0.003$, t=$-1.97$; F[1,56]=4.60; p=.037). In all four pregnancies, fetal orofacial movements increasingly match contact call temporal profiles as gestation progresses (*Figure 4C*; clockwise from top left: singleton, quadruplets, twins, and twins).

We further quantified the developmental change leading up to this match in a number of ways. We first looked at the durations of fetal orofacial movements over the course of gestation (n=1845;

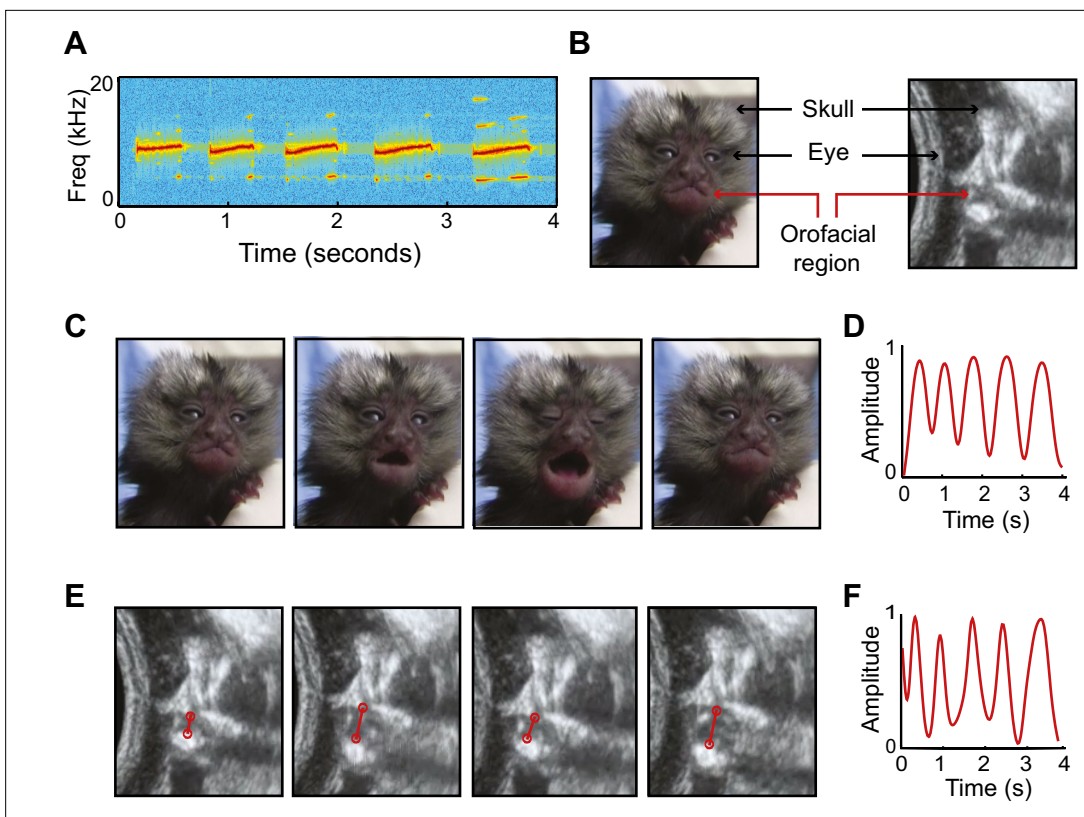

**Figure 3.** Signature features of marmoset infant calls observed in fetal movements. (**A**) Spectrogram of a five-syllable infant contact call from postnatal week 1. (**B**) Stills of marmoset infant and fetus, indicating orofacial region. (**C**) Video stills of week 1 infant producing a five-syllable contact call. (**D**) Temporal profile of the infant contact call in (**C**), generated by tracking orofacial movements. (**E**) Ultrasound stills of the orofacial movements of a late-stage marmoset fetus. (**F**) Temporal profile of the fetal orofacial movements in (**E**). The match between infant and fetal orofacial movements can be observed in *Video 3*.

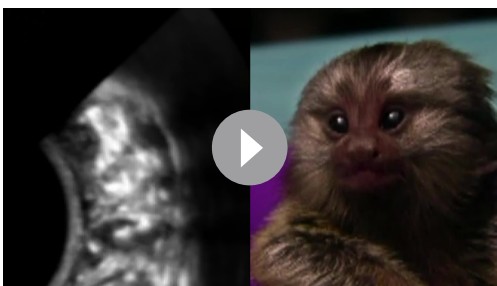

**Video 3.** Fetal orofacial movements matching the infant contact call.
https://elifesciences.org/articles/78485/figures#video3

*Figure 4D*). The median, 75th and 25th percentile durations were calculated for each session. To determine the developmental trajectory, the best polynomial-fit order using AIC was found to be three for the median values. The same polynomial order was used for generating curves for the 75th and 25th percentiles values. Kernel density estimation was used to visualize the duration density change over the course of gestation. The durations of orofacial movements show an increasing trend. Comparing the fetal orofacial movement durations to those of P1 infant contact calls, the fetal movements are seen to increasingly approach the P1 durations (*Figure 4E*; n=120; median duration = 3.69s; SE = 0.14).

Since contact calls typically contain multiple syllables, we calculated the 'syllable number' of the fetal orofacial movements (n=1845; *Figure 4F*). This is the number of individual orofacial movements separated by <500 ms (i.e. the same criteria used for counting actual contact call syllables in infants *Zhang and Ghazanfar, 2016*). The median, 75th and 25th percentile syllable numbers were calculated for each session. The best polynomial-fit order using AIC was found to be two for the median values. The same polynomial order was used to fit curves for the 75th and 25th percentiles values. The polynomial curves and kernel density estimation show that the syllable numbers of fetal orofacial movements increase through gestation. Fetal syllable numbers get closer to the syllable numbers of contact calls produced on P1 (*Figure 4G*; n=120; median = 5; SE = 0.18).

To directly compare the features of fetal orofacial movements with signature features of the infant contact call, we established criteria for identifying the contact call based on duration and syllable number. Contact calls tend to have long durations, and multiple syllables when compared to other infant calls. An infant call within the duration range of 3.69–6.5 s and syllable number range 5–9 has a 97% likelihood of being a contact call. To track the development of 'contact call-like' fetal orofacial movements, we calculated, for each session, the percentage of movements that matched the duration and syllable number features of P1 contact calls (*Figure 4H*). A multiple linear regression analysis controlling for the different pregnancies shows that fetal orofacial movements increasingly match the contact call profile as the fetus gets older ($\beta$±SE = 0.15±0.04, t=3.44; F(1, 63)=12.09, p=0.001). All four pregnancies were seen to have more 'contact call-like' movements over the course of gestation (*Figure 4I*; clockwise from top left: singleton, quadruplets, twins, twins). (Statistics for each individual pregnancies are available in *Supplementary file 1*).

To control for the possibility that the fetal orofacial movement profile is a generic one and not specifically linked to contact calls, we compared it to the two other orofacial movements common in the first weeks of postnatal life: twitter vocalizations and licking. When separated from caregivers, marmoset infants also produce bouts of another type of vocalization known as twitters. Relative to contact calls, twitters are short duration calls with fewer syllables. First, we checked if this contrasting type of vocalization produced by the P1 infants (n=109 twitters) follows a similar developmental trajectory. An infant call within the duration range of 0.96–1.5 s and syllable number range 2–3 has a 86% likelihood of being a twitter call (and a 2% likelihood of being a contact call). We then calculated, for each session, the percentage of fetal movements that matched the duration and syllable number features of P1 twitter calls (*Figure 5A*). A multiple linear regression analysis controlling for the different pregnancies reveals that—in contrast to what is seen with the contact call—fetal orofacial movements do not increasingly match the twitter profile as the fetus gets older ($\beta$±SE = 0.06±0.06, t=0.99; F(1, 63)=0.87, p=0.35).

We observed that when separated from their caregivers the infants often produced licking movements. (Note: weaning begins at ~30 days in marmosets, therefore chewing movements are not seen in neonates.) We measured the duration and syllable number features of these movements in the same infants, from ages P1-7 (n=37) and found that licks have greater variability than the contact call both in duration (*Figure 5B*; median = 1.6s; SE = 0.83) and 'syllable' number (number of jaw

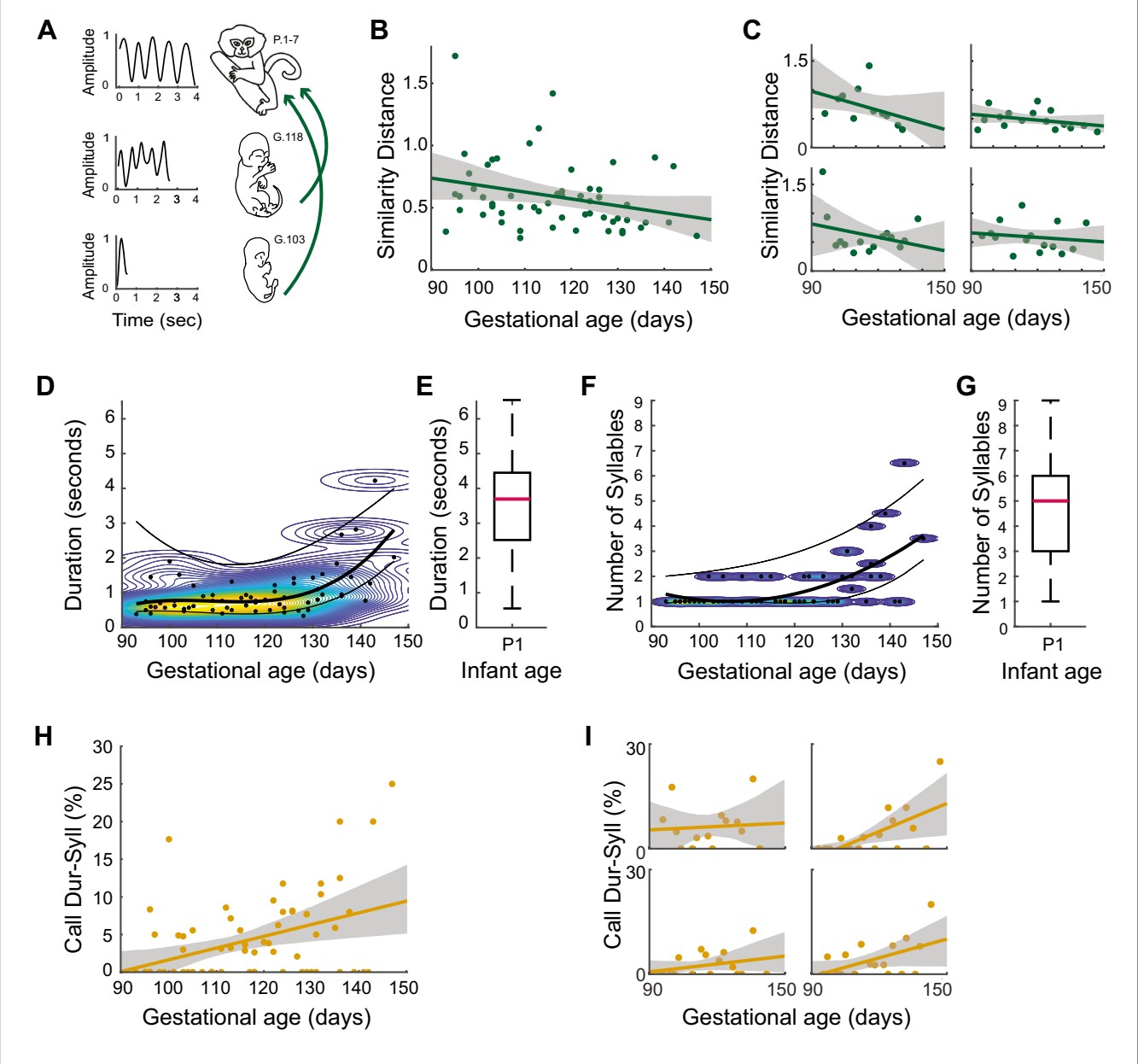

**Figure 4.** Signature features of marmoset infant calls on P1-7, emerge prenatally as distinct patterns of orofacial movements. (**A**) Illustration of the gestational change in the temporal profiles of fetal orofacial movements. On the left are the temporal profiles generated by tracking orofacial movements. On the right are the ages of the animals. (**B**) Matching temporal profiles of fetal orofacial movements to that of week 1 infant contact calls. Points represent the DTW similarity distance (lower the similarity distance, higher the matching), and the line represents the regression fit. ($\beta\pm$SE = −0.005±0.003, t=−1.97; F[1,56]=4.60; p=.037, multiple linear regression analysis controlling for the different pregnancies) (**C**) Same as (**B**), for each pregnancy. (**D**) Gestational change in the durations of fetal orofacial movements. Points represent the median movement duration per session (n=1845 total movements, 64 sessions). The curves are fit to the median, 75th and 25th percentile values. The curves were generated by calculating the optimal polynomial degree for the median values. The background contour map indicates durations with the highest density of movements (red: high density; blue: low density). (**E**) Infant contact call durations on P1. The pink bar marks the median contact call duration (n=120; median duration = 3.69s; SE = 0.14). (**F**) Gestational change in the syllable number of fetal orofacial movements. Points represent the median syllable number per session (n=1845 total movements, 64 sessions) . The curves are fit to the median, 75th and 25th percentile values. The curves were generated by first calculating the optimal polynomial degree for the median values. The background contour map indicates syllable numbers with the highest density of movements (yellow: high density; blue: low density). (**G**) Infant contact call syllable numbers on P1. The pink bar marks the median syllable number (n=120; median = 5 syllables; SE = 0.18). (**H**) Fetal orofacial movements matching infant contact call duration+syllable number signatures. Points represent the match percentage per session, and the line represents the regression fit ($\beta\pm$SE = 0.15±0.04, t=3.44; F(1, 63)=12.09, p=0.001, in the multiple linear regression analysis controlling for the different pregnancies). (**I**) Same as (**H**), for each pregnancy.

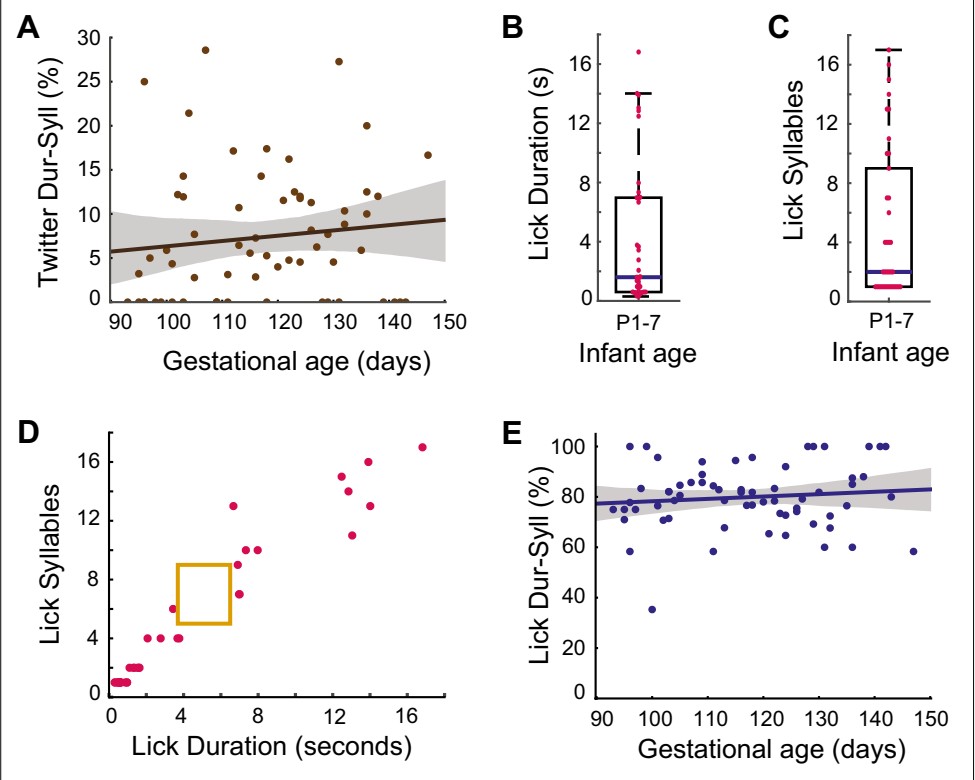

**Figure 5.** Prenatal developmental change specific to contact calls. (**A**) Fetal orofacial movements matching infant twitter call duration +syllable number signatures. Points represent the match percentage per session, and the line represents the regression fit ($\beta$±SE = 0.06±0.06, t=0.99; F(1, 63)=0.87, p=0.35, multiple linear regression analysis controlling for the different pregnancies). (**B**) Infant lick durations on P1-7. Points represent the durations of individual movements. The blue bar marks the median lick duration (median = 1.6s; SE = 0.83). (**C**) Infant lick call syllable numbers on P1-7. Points represent the syllable numbers of individual movements. The blue bar marks the median syllable number (median = 2; SE = 0.85). (**D**) Infant lick call duration +syllable number signatures distinct from contact calls. Points represent the duration and syllable number features of individual licks. The yellow box represents the contact call duration and syllable number profile. (**E**) Fetal orofacial movements matching infant lick call duration +syllable number signatures. Points represent the match percentage per session, and the line represents the regression fit ($\beta$±SE = 0.10±0.11, t=0.97; F(1, 63)=0.78, p=0.38, multiple linear regression analysis controlling for the different pregnancies).

movements) (*Figure 5C*; median = 2; SE = 0.85). We found a 0% likelihood that a movement within the duration range of 3.69–6.5 s and syllable number range of 5–9 (our contact call criteria) is a licking-related movement (*Figure 5D*). Infant licking movements are distinct from their contact calls.

We then tracked the development of 'lick-like' orofacial movements in the fetus. The licks of P1-7 infants tend to be short with few syllables, or very long with a high number of syllables. To capture the maximum possible infant licks, we established separate criteria for each of these types of licks: duration range of 0.33–3.77 s and syllable number range of 1–6 for the short licks and duration range of 6.67–14.29 s and syllable number range of 10–16 for the long licks. Together they capture 83.78% of the infant licks (and 0% of the infant contact calls). We then calculated, for each session, the percentage of fetal movements that matched the features of licks (*Figure 5E*). We identified more 'lick-like' orofacial movements than 'contact call-like' movements in the fetus, but a multiple linear regression analysis controlling for the different pregnancies reveals that fetal orofacial movements do not increasingly match the lick profile as the fetus gets older ($\beta$±SE = 0.10±0.11, t=0.97; F(1, 63)=0.78, p=0.38). These comparisons establish that the changes in movements we have tracked are specifically related to the rhythmic contact calls made by neonatal marmosets. It is worth noting that an increase in lick-like movements over the course of development could have been missed due to ceiling effects, i.e., they may have developed at an earlier time in gestation. Nevertheless, we can conclude is that in the gestation period under observation (~E93 to birth), there is no significant increase in lick-like

movements. However, as we hypothesized, the marmoset contact call production has a period of prenatal development.

## Discussion

The neonate's well-being hinges on using vocalizations to solicit attention. In human and non-human primates, these neonatal vocalizations transform into mature vocal signals through continuous and reciprocal interactions among the infant's advancing vocal apparatus, neural circuitry, and interactions with caregivers (*Ghazanfar and Liao, 2018*; *Teramoto et al., 2017*; *Thelen et al., 1991*). Our study focused on the prenatal origins of those very first vocalizations; particularly the development of their spatiotemporal signatures, which we tracked using vocalization-related mouth movements. Using fetal marmoset monkeys, we show that early mouth movements are tightly coupled to head movements. As gestation progresses, the movements decouple, ultimately leading to the formation of two independent movements. In parallel, a subset of fetal orofacial movements gradually assumes the pattern necessary to produce the immature-sounding contact calls neonates use to solicit care (*Huang et al., 2020*) and the parental interactions used to ratchet their vocal development (*Takahashi et al., 2015*; *Takahashi et al., 2017*; *Gultekin and Hage, 2017*; *Gultekin and Hage, 2018*; *Takahashi et al., 2016*). Aspects of the sensory-motor development necessary for vocalizing are occurring prenatally, even before the production of sound.

That body parts (head and mouth movements) were initially linked in marmoset fetuses is consistent with the developmental pattern of fetal limb movements across many species (*Hamburger, 1963*; *Barron, 1941*; *Coghill, 1929*; *Oppenheim, 1974*; *Robinson, 1988*). For example, early in gestation, embryonic chicks move their limbs and wings together. It is only in later prehatching life that these two body parts uncouple in their movements and can perform separate functions (*Bradley, 1999*). In sheep, the first limb movements are paired with head and neck motions, but these different body parts acquire increasing autonomy over the course of development (*Barron, 1941*). These fetal limb movements play an important role in the formation and organization of the spinal cord and CNS (*Granmo et al., 2008*). However, the development of these head and mouth action patterns in marmosets involve central pattern generators located in the brainstem not the spinal cord (*Barlow et al., 2010*; *Moore et al., 2014*). This suggests that the principles of fetal development—whereby linked movements involving multiple body parts are gradually unlinked—are general.

One question is whether the underlying mechanisms are the same for the patterns of change seen for head-mouth movement differentiation versus the differentiation between limb movements and other parts of the body. Studies of chicks suggest that the constraints of the egg help differentiate motor units and construct species-typical patterns of embryonic motility (*Bekoff, 2001*). Increasing uterine constraint has also been used to explain the common inversed-shaped pattern of limb movement frequency observed in many species (*Barron, 1941*; *Oppenheim, 1974*; *Robinson, 1988*; *Gottlieb, 1976*; *Roodenburg et al., 1991*). In our study of marmosets and in humans (*Roodenburg et al., 1991*), the frequency of mouth movements over the course of gestation also showed an inversed U-shaped trajectory. Do they decrease in number because, as the fetus grows larger, there is less room to make orofacial movements? This seems unlikely in our study. In the four pregnancies we monitored, the number of fetuses (and thus the space constraints) varied, but there was no indication that there was any related difference in the developmental trajectory of mouth movements. Consistent with this idea, a study of fetal rats compared their activity levels under three different, decreasing space constraints: the uterine environment, outside the uterus but with extra-embryonic membranes intact, and with all membranes removed (*Smotherman and Robinson, 1988*). In all cases, activity profiles still exhibited the familiar inversed U-shaped profile (*Smotherman and Robinson, 1988*). These data suggest that the shape of the developmental trajectory is influenced by neural changes, possibly from the developing forebrain (*Barron, 1941*; *Decker and Hamburger, 1967*). Indeed, a computational model of motor development supports the idea that spontaneous motor activity can self-organize into reflex actions, and that these actions are subsequently modulated to produced coordinated, adaptive behaviors (*Marques et al., 2014*).

In utero interactions between the species-typical body and brain, and between the body and its environment are hypothesized to drive the development of muscle and neural organization, and, ultimately, lead to the development of species-typical behavior (*Bekoff, 2001*). This idea is supported by computer simulations of fetal development that model fetal motor behavior and spinal circuit changes

of different mammalian species within a uterine environment (*Gottlieb, 1976*; *Roodenburg et al., 1991*; *Decker and Hamburger, 1967*; *Marques et al., 2014*). At beginning of simulations, all fetuses had different musculoskeletal bodies but the same undifferentiated neural circuits. Following spontaneous activity, the simulated fetuses developed species typical motor patterns, e.g., the human fetus developed alternating leg movements and the zebrafish embryo developed coordinated side-to-side movements. A by-product of these simulations was that individual muscle contractions were initially part of larger undifferentiated motor units (e.g. a single unit controlling the head and the trunk), but as development progressed, these large muscle units subdivided into smaller, more precisely controlled units (e.g. separate control of the head and the trunk). Our finding that head and mouth motor units are linked early in gestation and uncouple with age provide additional support for this hypothesis.

A subset of fetal marmoset mouth movements were precursors to the movements needed to produce contact calls as neonates; over gestation, they became increasingly similar in their temporal structure to neonatal articulatory movements. Likewise, the movement patterns of fetal rodent limbs are seen to get quantitatively and qualitatively closer to those of the neonate (*Brumley and Robinson, 2010*; *Kleven et al., 2004*; *Robinson and Kleven GA, 2005*). These same phenomena are observed in chicken embryos (*Bradley and Sebelski, 2000*; *Sharp et al., 1999*). These studies in rodents and birds suggest that experiences in-utero play a key role in establishing early postnatal movements. For example, fetal rats modify their limb movements in response to physical constraints, exhibiting motor learning in utero (*Robinson and Kleven GA, 2005*). The simulations described above also support the notion that in utero learning via sensory feedback generated by fetal movements lead to species-typical stepping or swimming actions (*Kuniyoshi, 2019*; *Mori and Kuniyoshi, 2010*; *Yamada et al., 2010*; *Yamada and Kuniyoshi, 2012*; *Yamada et al., 2016*; *Mori, 2012*). Thus, it does not seem too far-fetched to suggest that self-generated experience leads to the convergence of the temporal structure of marmoset fetal mouth movements toward the pattern of neonatal vocal articulatory movements (at least for contact calls). Self-generated learning of this type would be of the same kind that is observed postnatally in mammals, whereby myoclonic twitches during sleep help construct sensorimotor maps in the brain (*Dooley and Blumberg, 2018*; *Dooley et al., 2020*; *Blumberg et al., 2013a*) (for review, see *Blumberg et al., 2013b*).

Here are some caveats to our study. Our study focused only on contact calls because they are among the first vocalizations produced by infants (albeit immature sounding) and are important for soliciting both vocal responses (*Takahashi et al., 2016*) and retrieval from caregivers (*Huang et al., 2020*). The most obvious parallel to marmoset monkey contact calling (and likely all infant contact calling in mammals and birds) is crying by human infants. The comparison, however, is not so simple. The contact vocalizations that infant marmosets use to solicit caregiver attention are also used by the adults to maintain social contact with conspecifics; this type of vocal behavior seems to be true for all nonhuman primates (*Newman, 2004*). In humans, communication with caregivers through crying soon gives way to other types of vocalizations and eventually speech (*Oller, 2000*). Next, in pregnancies with multiple fetuses, we treated them as a single composite subject. This was because there was no way to identify the same individuals across days. Two findings mitigate against this issue: all four pregnancies exhibited the same behavioral profiles and developmental trajectories, and the early postnatal vocal output of twins are nearly identical in their patterns of output (more so than in comparison to their non-twin siblings), seemingly driven by arousal levels regulated by their caregivers (*Zhang and Ghazanfar, 2016*).

Another caveat is that while we implicitly interpreted the fetal orofacial movements we observed as being 'spontaneous' as other similar studies have, it is possible that these movements could have been initiated by activation of the developing sensory system (*Hamburger, 1963*). However, we did not observe any correlation with, e.g., application of the ultrasound probe and fetal movements, and a pilot study by us to elicit fetal movements via sound playback near the mother's belly was unsuccessful. Relatedly, our previous work (*Zhang and Ghazanfar, 2016*) showed the initiation and duration of infant vocal output is linked to a 0.1 Hz autonomic rhythm and that the temporal structure of those sequences are correlated with the respiratory cycle. Thus, one possibility is that the timing of fetal bodily and/or orofacial movements are also tied to such cycles, or at least to the autonomic rhythm (in humans at least, fetal 'practice' breathing movements are intermittent). Early on in our study, we attempted to investigate this link but we could not simultaneously monitor heart rate (in mothers or fetuses) while also monitoring orofacial movements. Finally, our sample size consisted of four

pregnancies across two unrelated mothers. The imaging and analysis were very time intensive, and we opted to longitudinally sample individual pregnancies very densely as opposed to increasing the number of subjects. This allowed us to better map the shape of developmental trajectories (*Adolph et al., 2008*).

Marmoset monkeys offer a special opportunity for comparative investigations of vocal development with humans. Like humans, marmosets exhibit a high degree of prosociality (*Burkart et al., 2014*) (and related phenotypes linked to 'self-domestication' [*Ghazanfar et al., 2020*]), show context-dependent vocal control and plasticity in the timing, intensity and spectral features of their vocalizations (*Choi et al., 2015*; *Takahashi et al., 2013*; *Zürcher et al., 2019*; *Liao et al., 2018*), and infant marmosets undergo a period of vocal learning that is influenced by parents (*Takahashi et al., 2015*; *Takahashi et al., 2017*; *Gultekin and Hage, 2017*; *Gultekin and Hage, 2018*). All of these similarities may be related to the fact that both humans and marmosets are cooperative breeders and are born altricial relative to other primates (*Varella and Ghazanfar, 2021*). Thus, we predict that the prenatal patterns of vocalization-related orofacial movements in marmosets we observed here is of the same kind in humans. This would open up possibilities to use marmoset monkeys to investigate how very early prenatal orofacial motor behaviors can be diagnostic of later vocal behaviors and prelinguistic vocal learning.

## Materials and methods
### Subjects
The subjects used in this study were nine fetuses from four pregnancies; three pregnancies of the same adult female, and one pregnancy of a different adult female. The two pregnant females came from different social groups and lineages. Of the four pregnancies, one was of a singleton, two of twins and one of quadruplets. Of the nine fetuses, seven survived (two of the quadruplets were stillborn). There were no obvious differences in the sizes of the singleton, twins, or quadruplets at birth. Our veterinary doctor performed necropsies on the stillborn animals and concluded that both were well developed and looked to be at term. The seven surviving fetuses were then tested as infants between postnatal days 1 and 7. The colony room was maintained at a temperature of ~27°C with 50–60% relative humidity and a 12:12 light/dark cycle. The animals had ad libitum access to water and were fed daily with a commercial marmoset diet, supplemented with fresh fruits, vegetables, and insects. All experiments were performed with the approval of the Princeton University Institutional Animal Care and Use Committee (protocol #1908–18).

### Experimental setup–ultrasonography
Ultrasonography tests were performed using a GE Voluson i Ultrasound machine. The testing method used was adapted from the procedure developed by S.D. Tardif and colleagues (*Jaquish et al., 1995*). Each examination was conducted by two experimenters—one gently restrained the animal while the other carried out the ultrasound procedure. The animals were not anesthetized for the examination. Instead, they were trained to accept restraint by hand and the gentle placement of the probe on their stomach. They were rewarded with treats at the end of the testing period. We routinely scanned all our adult females for pregnancy. Early pregnancy was detected by assessing the morphology of the uterus (*Jaquish et al., 1995*). Once pregnancy was confirmed, ultrasounds were conducted every 2 weeks to monitor the development of the fetus. When the fetal skull became clearly visible, gestational age could be estimated by measuring the biparietal diameter of the skull (*Jaquish et al., 1995*). At this point, we started monitoring the fetus 2–3 times a week. As soon as the fetal face was clearly visible (~95 days gestational age), the ultrasound imaging was fixed on the face as much as possible, and recordings were taken for 15–45 min every other day (by this time the mothers had become accustomed to the procedure, and we saw no evidence in their behavior that their stress levels systematically increased through the gestation period). In cases where there was more than one fetus, two fetuses were selected at random, and each fetus was observed for roughly half the time. We differentiated the fetus by its position in the womb during the session. The examination was terminated if the animal showed significant resistance, hence the variability in session length. All examinations were conducted between 1400 and 1800 hr, and the procedure was repeated using exactly the same procedure until birth (~146 days gestational age). The ultrasound videos were captured at a

frame rate of 30 Hz and written on to DVD for later analysis. A total of 64 sessions were recorded—14 sessions for the singleton, 17 sessions for the first set of twins and the quadruplets, and 16 sessions for the second set of twins.

## Scoring videos

Video analysis was done using Adobe Premiere Pro software, which enabled frame-by-frame analysis of the ultrasound movement segments. The ultrasound still images are visually similar to x-ray images. The bones of the face are seen in white, while the mouth and eye cavities appear dark in contrast. To mitigate against scoring bias, scoring was done blind to gestational age. The videos were not tagged with gestational age and an accurate gestational age of the fetus cannot be determined by eye alone since the ultrasound probe placement (angle, distance from the fetus etc.) can change the apparent size of the fetus. Only a measure of the biparietal distance of the fetal skull provides an accurate estimate of age. Moreover, videos were scored and analyzed long after (several months after) they were acquired. The analyses of these scores were done long after (several months after) the scoring. Finally, the videos were scored by three individuals, with every alternate day being scored by a different person. Two of the three scorers were not involved in formulating the hypothesis or performing data analyses.

To reduce experimenter error, we developed explicit criteria for scoring the videos, and roughly half of the total 64 sessions were coded by one experimenter, and the other half by another (every other session done by the same person). Randomly chosen recording session videos were coded by both experimenters to check for coding consistency. The marmoset mothers were not anesthetized for the ultrasound procedure, and both the mother and the fetus could move, causing us to lose sight of the fetal face from time to time. Therefore, we first identified segments of the video during which the fetal face was clearly visible. Within these segments, we scored for orofacial and head movements.

## Identifying orofacial and head movements

A movement counted as an orofacial movement when there was a clear separation of the upper and lower jaws. The first video frame where the jaws separated, and the dark region between the upper and the lower jaws first began to increase, counted as the beginning of orofacial movement. The video frame where the jaws fully came back together was the end point of the movement. If we lost focus of the face before the jaws came back together, the last frame of observation counted as the end of the movement. We did not consider the surface area of mouth opening or the amplitude of the orofacial movements in our analysis as those measures are sensitive to the position and orientation of the moving fetus. We therefore only considered the relative position of the upper and the lower jaws, which is easily and consistently discernible, even in fetuses as young as E90.

Individual orofacial movements were considered to be part of the same movement unit if they were separated by 500 ms or less (15 frames). The 500 ms criterion was justified by the bimodal structure of the inter-syllable interval distribution of the vocal output of marmoset neonates. In their calls, the 500 ms threshold separates the first mode of the distribution (representing the interval between syllables within a single call) from the second mode that represents the interval between the offset of the last and onset of the first syllables between two calls (*Zhang and Ghazanfar, 2016*). We made a note of partially captured movement units. These units were included when counting the number of movement units, as in *Figure 2*. We excluded the partially captured movement units when calculating duration and syllable number, as in *Figure 4*, since the fetus moved out of focus while the mouth was still open.

Similar to orofacial movements, the first video frame where the head moved away from baseline position was the beginning of movement, and the first video frame where the head came back to baseline was the end point of movement. The baseline position of the head was determined using the region where the upper and lower jaw meet, so as not to conflate head movements with the movement of the lower jaw. If we lost focus of the fetus before the head came back to baseline position, the last frame of observation counted as the end of the movement. We excluded the partially captured head movements when calculating the duration of movements. We applied the same criteria used for the orofacial movements, when combining individual head movements into a single unit. In the article, when we refer to orofacial or head movements, we are speaking of movement units.

In young fetuses, where the body can be seen in addition to the head, other body parts were often moved with the head. We decided to focus on the head movements alone since the head region can be consistently observed with the orofacial region, even in the older fetuses where the body and the orofacial region cannot be imaged simultaneously. In addition, simulation experiments show that head and orofacial motor units are some of the last to differentiate (*Yamada and Kuniyoshi, 2012*).

## Exemplars of fetal orofacial and head movements

For the movement exemplars, the ultrasonography still images were generated first by using a custom-made MATLAB program to split a chosen video clip into its component frames, and then selecting those frames which best exemplified the movement. The movement timeline plot was generated using onset and offset information of the movements (method described above) and plotted using MATLAB. The amplitude of the movement did not have any bearing on the analysis.

## Information theory analysis—testing for increase in 'order' through gestation

In *Figure 2A–D*, each movement occurrence was assigned to one of 5 states:

> State 1: independent orofacial movement.
> State 2: independent head movement.
> State 3: orofacial movement followed by an overlapping head movement—if an orofacial movement was initiated at least one frame (~30 ms) after the beginning of a head movement and before the end of the movement.
> State 4: head movement followed by an overlapping orofacial movement—if a head movement was initiated at least one frame (~30 ms) after the beginning of an orofacial movement and before the end of the movement.
> State 5: orofacial plus head movements with synchronous onset (movements initiated in the same frame).

A first order Markov model was used to determine the state transitions (*Figure 2B*).
The state distribution of each session was determined.
Shannon Entropy (H) for each session was calculated using the following formula:

$$H(X) = -\sum_{i=1}^{5} P\left(X = i\right) \log P\left(X = i\right)$$

where P(X=i) for i=1–5 (corresponding to the 5 states) is the frequency of occurrence of each state. We use the convention 0 log 0=0, when a state probability is null (*Figure 2C*).
Kullback-Leibler Divergence of Q from P was calculated using the formula:

$$KL\left(P|Q\right) = \sum_{i=1}^{5} P\left(X = i\right) \log P\left(X = i\right) / Q\left(X = i\right),$$

where i indicates different states and log is in base 2; P was the average state distribution for the first 6 testing sessions (gestational day 93–99 days, across all four pregnancies); Q was the state distribution for every testing session (gestational days 93, 95….147) (*Figure 2D*).

## Developmental change in the rates of orofacial and head movements

The numbers of orofacial and head movements were counted for each session. Here too, roughly half of the total 64 sessions were coded by one experimenter and the other half by another (alternate days done by the same person). Since the amount of time the face was visible differed between sessions, we calculated the rate of orofacial and head movements per hour. Movement rates were compiled across all pregnancies and polynomial curves (one for orofacial movements and one for head movements) were fitted to look at the trends across gestational time. To fit the curves, we first found the optimal degrees for polynomial fitting according to AIC (*Akaike, 1981*)(MATLAB polydeg). These degrees were then used in a polynomial curve fitting function (MATLAB polyfit) to generate optimal fits for orofacial and head movement rates. To check if the observed trends held for individual pregnancies, we split the orofacial and head movement rates by pregnancy and used the optimal polynomial degrees calculated with the entire dataset to generate polynomial curves. The profiles of orofacial and head movement rates were found to be very robust, and the same profiles were seen when the analysis was repeated by looking at movement durations instead of movement numbers.

### Developmental change in the overlap of orofacial and head movements

We counted the number of instances where orofacial and head movements occurred together and divided this number by the total number of orofacial and head movements (the same trend was found if the duration of overlap was used instead). The overlap calculation included instances where orofacial movements were followed by an overlapping head movement, instances where head movements were followed by an overlapping orofacial movement and instances where orofacial and head movements started simultaneously. To determine the trend across gestational time, we first found the optimal degrees for polynomial fitting according to AIC (*Akaike, 1981*) (MATLAB polydeg). This degree was then used in a polynomial curve fitting function (MATLAB polyfit) to generate the curve. To check if the observed trends held at the pregnancy level, we split the overlap rates by pregnancy and used the optimal polynomial degrees calculated with the entire dataset to generate polynomial curves. To test for the significance of decrease in the overlap rate, we fitted a linear regression between the overlap rate and gestational date controlling for the pregnancies and tested for the nullity of the effect (ANOVA) of the gestational date.

### Permutation test

The total number of orofacial and head movements was seen to decrease through gestation. To confirm that the observed decline in orofacial and head movement overlap was independent of the general decline in movement, a permutation test was performed. To do this, we took each session and independently shuffled both the orofacial and head movements, keeping the duration and latency distributions intact. We then independently resampled with replacement, keeping the number of durations and intervals the same as the original dataset. Overlap between orofacial and head movements was then calculated. This was done for every session. Average percentage of overlap was calculated for each gestational age (same procedure as that used in our original calculation). This procedure was repeated 1000 times and the upper 97.5 and lower 2.5 percentiles were computed to generate the confidence interval. To fit the curve, we followed the same procedure as with the observed data—we first found the optimal degrees for polynomial fitting according to AIC (*Akaike, 1981*) (MATLAB polydeg). This degree was then used in a polynomial curve fitting function (MATLAB polyfit) to generate the curve. We expected this analysis to show the pattern of overlapping movements across gestation that would have resulted due to chance.

### Change in durations of orofacial and head movements through gestation

Orofacial and head movement durations were calculated by subtracting a movement's offset time from its onset time. Movements that were marked as 'partially captured' during the initial scoring were excluded from this analysis. Median orofacial and head movement durations for each session were calculated and compiled across all sessions, for all pregnancies. To test if there was a change in the duration of orofacial and head movement through gestation, we used the same curve fitting method used for movement and overlap rates.

### Experimental setup—infant vocalizations

Audio and video recordings of infant vocalizations were made from P1 (~24 hr after birth) to P7. The infant was briefly separated from its caregivers and taken to a 2.5 m × 2.5 m room with walls covered in sound attenuating foam. For the audio recordings, the infant was placed on a layer of foam in a transfer cage. Once the subject was in place, the experimenter turned on a digital recorder (ZOOM H4n Handy Recorder) positioned directly in front of the testing cage at a distance of 0.76 m and left the room for a period of 5 min. For the video recordings one experimenter held the baby and a second experimenter acquired videos of the infant face using a hand-held SONY video recorder. This was done to observe facial movements during vocalizations. The frame rate of the infant videos matched that of the ultrasound videos—both were 30 frames per second.

### Infant contact calls—audio and video recordings

Audio recordings were processed using Adobe Audition software. The spectrograms of the audio signals were used to identify the infant contact call. In marking the calls, onset-offset gaps of 500

ms or longer indicated separate calls, whereas gaps <500 ms indicated syllables from the same call (*Zhang and Ghazanfar, 2016*).

Video analysis was done using Adobe Premiere Pro software, which enabled us to screen the videos frame-by-frame. The exact same frame-by-frame analysis used to score the fetal videos was used for the infants. The first video frame where the jaws separated counted as the onset of orofacial movement and the first video frame where the jaws came back together was the offset of the movement. Here too, individual orofacial movements were considered to be part of the same movement unit, if they were separated by <500 ms (i.e.<15 frames) (*Zhang and Ghazanfar, 2016*).

For our analysis, we used the P1 audio recordings and P1-7 video recordings. Marmoset infants frequently fall asleep on day 1 (just like any other newborn), especially when held. Therefore, to get a large enough sample size, we used the first week of video recording. The audio recordings on P1 consisted of a total of 120 contact calls, the video recordings from P1-7 consisted of a total of 42 contact calls. We combined both data to obtain enough samples.

To compare the audio and video signals, we extracted audio from the video recordings and measured call duration and syllable number separately in each modality. We then measured the difference in duration and syllable number in the two signals. In case of duration, the video measures were consistently longer than the audio. To compensate for the duration difference, we calculated the difference between the video and audio duration. The median duration discrepancy was added to the audio duration of all contact calls to adjust for the discrepancy. The number of syllables remained the same across the audio and video measures; no adjustment was required.

## Infant contact call exemplar

The image panels were generated by first using a custom-made MATLAB program to split the chosen infant contact call video clip into its component frames and then selecting those frames which best exemplified the movement.

The spectrogram was produced by extracting the contact call audio signal from the chosen clip and reducing background noise using the Adobe Audition software.

The temporal profiles of the accompanying orofacial movements were generated using a custom MATLAB program which allowed us to go through the movement clip frame-by-frame and mark the positions of the lower and the upper jaws by selecting a central point on each jaw. (When there is no orofacial movement, the distance between the two points is 0, while a nonzero value indicates that the mouth is open.) The profiles were then z-scored to remove amplitude information. The generated profiles were smoothed using the cubic smoothing spline (csaps) function in MATLAB (smoothing parameter = 0.999).

## Fetal contact call precursor exemplar

Temporal profiles of a set of fetal orofacial movements that were similar in duration and syllable number to the infant contact call were generated using the same custom MATLAB program and procedure used for generating infant contact call profiles. A profile that best matched the infant contact call profile was selected as the exemplar.

## Matching temporal profiles of fetal orofacial movements to that of week 1 infant contact calls

Videos of P1-7 infants producing contact calls were compiled (n=25), and a custom MATLAB program was used to do a frame-by-frame tracking of the upper and lower jaw positions to create temporal profiles for the contact calls. These calls consisted of 1–7 syllables (1 syllable, n=2; 2 syllables, n=3; 3 syllables, n=5; 4 syllables, n=5; 5 syllables, n=2; 6 syllables, n=4; 7 syllable, n=4). The profiles generated from calls of the same syllable number were averaged using the DTW Barycenter Averaging (DBA) MATLAB routine (*Petitjean et al., 2011*). The average profiles were then smoothed using MATLAB spline smoothing function csaps (smoothing parameter = 0.1), to generate the templates of infant contact calls (n=7).

A custom computer vision program was developed in C++ to track fetal orofacial movements in an automated fashion. The program was set up with a MATLAB interface. Through this interface, we loaded individual movement clips, and marked the head, nose, and orofacial regions of the fetus on

the first frame of the clip. The program then went through the rest of the clip frame-by-frame and generated the temporal profile of the orofacial movement.

Fetuses move significantly in the womb, especially in the mid stages of gestation. To make sure we only included accurate tracings of the movements, we scored the temporal profiles of all the movements (n=1977). All the partially captured movements were excluded, and only traced profiles with highest score—those that matched the manually calculated syllable number features of the movement—were included in the rest of the analysis (n=460).

The temporal profiles were then smoothed using the Savitzky-Golay Smoothing Filter. The tuning parameters we used were: polynomial degree = 3 and sliding window size = 9. The tuning parameters were selected based on what retained the structure of the tracings and could be used for the majority of movements (n=414).

### DTW analysis

To measure the similarity between the two temporal sequences (fetal orofacial movement and infant contact call template) with possibly different time lengths, we used the continuous DTW algorithm using a linear interpolation model. We used the DTW Python module generated by Pierre Rouanet (version 1.3.3). The cost of the DTW was used as a measure of similarity between two signals. Smaller values of DTW cost indicate larger similarity between the signals. Each fetal movement profile was compared with all seven infant contact call templates, and a DTW score generated. The lowest DTW score was retained. DTW scores from all the movements in a single session were compiled and the median DTW score calculated. Median DTW scores were generated for all sessions (n=57). The statistical computing was done using R. We fitted a multiple linear regression model to predict similarity to the infant contact call temporal profile (DTW score) based on gestational age and pregnancy.

### Gestational change in the duration and 'syllable numbers' of fetal orofacial movements

Orofacial movement durations were calculated by subtracting a movement's offset time from its onset time. Syllable numbers were calculated by counting the number of mouth movements in each movement unit. Partially captured movements were not included in this analysis. Median, 75th and 25th percentile values for duration and number of individual movements in a movement unit (equivalent of syllable number) were calculated for each session. The developmental trajectories of duration and syllable number were generated by first calculating the optimal degree for polynomial fitting according to AIC (*Akaike, 1981*) (MATLAB polydeg), for the median values. The same polynomial fitting degree was used for the other two curves. The contour plots for median durations and syllable numbers were made using MATLAB routine 'Kernel density estimation' by Zdravko Botev (MATLAB file exchange #17204).

### Duration and syllable number features of the infant contact calls at P1

The boxplots for P1 infant contact call duration and syllable number distributions were done using the 'Box and whiskers plot (without statistics toolbox)' by Jonathan C.Lansey (MATLAB file exchange #42470). These plots indicate the median, 75th and 25th percentile values.

### Criteria for defining contact calls based on duration and syllable number features

To be able to match the durations and syllable numbers of fetal orofacial movements to the infant contact call, we had to obtain criteria based on orofacial movement that allowed us to select contact calls with high probability. Following is the method we used for setting the bounds. (1) For every call duration (500 ms–6.5 s) or syllable number (1–9), we calculated the likelihood that the call is a contact call based on our infant contact call samples (e.g. if the duration of a call is 6 s the likelihood that it is a contact call is 100%) (2) We set the duration and syllable number range for contact calls to ensure maximum possible separation from other call types. The duration criteria we arrived at for the infant contact call is 3.69–6.5 s. The syllable number criteria we arrived at is 5–9. When both the duration and syllable number criteria are applied, the contact call likelihood is 97%.

### Matching fetal orofacial movements to contact call criteria

The percentage of fetal orofacial movements matching the contact call duration +syllable number criteria ('percent match contact call dur-syll') was calculated for each session. The statistics were

calculated using R. We fitted a multiple linear regression model to predict percentage match of contact call dur-syll based on gestational age and pregnancy. ANOVA test was used to test for the significance of the effect of gestational age.

## Criteria for defining twitter calls based on duration and syllable number features

The criteria for defining twitter calls was obtained using the same method as that used for the contact calls. The duration criteria we arrived at for the infant twitter call is 0.96–1.5 s. The syllable number criteria we arrived at is 2–3. A call within this duration and syllable number criteria has an 86% likelihood of being a twitter call and a 2% likelihood of being a contact call.

## Matching fetal orofacial movements to twitter call criteria

The percentage of fetal orofacial movements matching the twitter duration+syllable number criteria ('percent match twitter dur-syll') was calculated for each session. The statistics were calculated using R. We fitted a multiple linear regression model to predict the percentage match of twitter dur-syll based on gestational age and pregnancy. ANOVA test was used to test for the significance of the effect of gestational age.

## Criteria for defining infant licks based on duration and syllable number features

Video analysis was done using Adobe Premiere Pro software, which enabled us to screen the videos frame-by-frame. The exact same frame-by-frame analysis used to score the fetal videos and infant vocalizations was used for the licking movements. The first video frame where the jaws separated was counted as the onset of orofacial movement and the first video frame where the jaws came back together was the offset of the movement. Here too, individual orofacial movements were considered to be part of the same movement unit if they were separated by <500 ms (i.e.<15 frames) (*Zhang and Ghazanfar, 2016*). Orofacial movement durations were calculated by subtracting a movement's offset time from its onset time. Syllable numbers were calculated by counting the number of mouth movements in each movement unit.

The licks of P1-7 infants (n=37) tend to be short with few syllables, or very long with a high number of syllables. Therefore, to capture the maximum possible infant licks, we established a separate criteria for each of these types of licks. Short licks: duration range 0.33–3.77 s and syllable number range 1–6. Long licks: duration range 6.67–14.29 s and syllable number range 10–16. Following is the method we used for determining the bounds: 0.33 and 14.29 are the 2.5 and 97.5 percentiles of the lick durations; while 1 and 16 are the 2.5 and 97.5 percentiles of the lick syllable number. There is only one movement of duration between the 3.77 and 6.67 s (therefore our duration criteria only filtered out 0.03% of the total movements). There are three movements of syllable number between 6 and 10 (therefore our syllable number criteria only filtered out 0.08% of the movements). Together our criteria capture 83.78% of the infant licks (and 0% of the infant contact calls).

## Matching fetal orofacial movements to lick call criteria

The percentage of fetal orofacial movements matching the lick duration +syllable number criteria ('percent match lick dur-syll') was calculated for each session. The statistics were calculated using R. We fitted a multiple linear regression model to predict the percentage match of lick dur-syll based on gestational age and pregnancy. ANOVA test was used to test for the significance of the effect of gestational age.

# Additional information

## Funding

| Funder | Grant reference number | Author |
| --- | --- | --- |
| National Institute of Neurological Disorders and Stroke | R01NS054898 | Asif A Ghazanfar |

The funders had no role in study design, data collection and interpretation, or the decision to submit the work for publication.

## Author contributions

Darshana Z Narayanan, Conceptualization, Formal analysis, Investigation, Visualization, Writing – original draft, Writing – review and editing; Daniel Y Takahashi, Conceptualization, Data curation, Formal analysis, Supervision, Validation, Investigation, Writing – review and editing; Lauren M Kelly, Formal analysis, Writing – review and editing; Sabina I Hlavaty, Formal analysis, Investigation; Junzhou Huang, Software, Methodology; Asif A Ghazanfar, Conceptualization, Supervision, Funding acquisition, Writing – original draft, Writing – review and editing

## Author ORCIDs

Darshana Z Narayanan http://orcid.org/0000-0003-3553-1875
Daniel Y Takahashi http://orcid.org/0000-0003-4972-001X
Asif A Ghazanfar http://orcid.org/0000-0003-1960-7470

## Ethics

This study was performed in strict accordance with the recommendations in the Guide for the Care and Use of Laboratory Animals of the National Institutes of Health. All of the animals were handled according to approved institutional animal care and use committee (IACUC) protocols (#1908-18) of Princeton University.

## Decision letter and Author response

Decision letter https://doi.org/10.7554/eLife.78485.sa1
Author response https://doi.org/10.7554/eLife.78485.sa2

# Additional files

## Supplementary files

• Supplementary file 1. Statistics for each individual pregnancies for the case of matching spatiotemporal profiles of fetal and neonatal orofacial movements.

• MDAR checklist

## Data availability

All data generated or analysed during this study are available on DRYAD https://doi.org/10.5061/dryad.m905qfv1x.

The following dataset was generated:

| Author(s) | Year | Dataset title | Dataset URL | Database and Identifier |
| --- | --- | --- | --- | --- |
| Ghazanfar AA | 2022 | Data from: Prenatal development of neonatal vocalizations | https://doi.org/10.5061/dryad.m905qfv1x | Dryad Digital Repository, 10.5061/dryad.m905qfv1x |

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
