## [Editor Report]

This paper will be of great interest to the field of developmental neuroscience and social communication. The authors were able to identify sensorimotor vocal precursors in fetal marmoset monkeys by using ultrasound imaging to detect rhythmic orofacial movements related to vocalizations. These findings provide new insights into the prenatal development of vocal behavior in primates. The data acquired by a highly quantitative approach support the major claims of the paper.

---

## [Decision Letter]

**Decision letter after peer review:**

Thank you for submitting your article "Prenatal development of neonatal vocalizations" for consideration by *eLife*. Your article has been reviewed by 3 peer reviewers, and the evaluation has been overseen by a Reviewing Editor and Andrew King as the Senior Editor. The following individuals involved in the review of your submission have agreed to reveal their identity: Yves Boubenec (Reviewer #1); Neng Gong (Reviewer #2).

Essential revisions:

The reviewers are generally very positive in their appraisal of your paper but have requested some additional analyses and clarification, which need to be included when submitting a revision of this paper.

1. Can you demonstrate that the manual image processing is not biased since this was not performed by individuals blind to the experiment who were likely, not unfamiliar with the different gestation stages in marmoset development?

2. The text is insufficiently clear in defining how state transitions are defined. Please clarify whether Shannon entropy and the KL divergence are just used for quantifying the relative proportion of each state rather than for quantifying the state changes per se. Further details of how the information theory analysis is affected by state probability are needed.

3. The criteria for defining the timing of head and orofacial movements were felt to be too rigid, so some exploration of this was felt to be desirable.

4. The KL divergence is quantified with respect to the first post-natal day. This may make this measure quite noisy. The effects of averaging over the first few days should be considered.

5. Please consider whether it is possible that an increase in lick-like movements over the course of development could have been missed because of a ceiling effect.

6. While we appreciate that you have attempted to show that orofacial movements are specifically related to vocal production, it was felt that this point needs to be more nuanced. Specifically, orofacial movements may be also important for sucking or breathing and are therefore part of the overall development of bodily movements. Furthermore, previous studies have shown that marmoset monkeys produce contact calls around every 10 s (~0.1 Hz), which are correlated with heart rate and "Mayer wave", an oscillation of the autonomic nervous system. Please address the concern that the link between orofacial movements and the production of contact calls could be explained by autonomic responses, such as breathing.

7. If possible, please comment on whether there is any evidence for the production of "cry" calls, similar to those of human babies, during the prenatal stage.

8. The apparent inconsistency with your previous work over whether marmosets produce contact calls immediately after birth needs to be addressed (see reviewer 3 comments).

9. The statistical results should be shown for the distributions of each monkey, to demonstrate how reliable these correlations are for each individual.

*Reviewer #2 (Recommendations for the authors):*

This is an interesting study, and the manuscript is well written. As described in the public review, my major concern is the overemphasis on the specific correlation between orofacial movements and vocal production (contact call). The authors may reconsider the interpretation and discussion of the data.

*Reviewer #3 (Recommendations for the authors):*

This is important work, and *eLife* is the right journal for these findings.

I have already seen an earlier version of this manuscript during the review process in another journal and am very pleased that the authors addressed most of my comments that I had at that time, such as the lack of comparison of the observed rhythmic oral movements with other rhythmic orofacial movements present from birth. The introduction of the controls performed, which clearly show that the prenatally observed rhythmic orofacial movements best match the infant contact calls, has significantly strengthened the manuscript.

---

## [Author Response]

Essential revisions:The reviewers are generally very positive in their appraisal of your paper but have requested some additional analyses and clarification, which need to be included when submitting a revision of this paper.1. Can you demonstrate that the manual image processing is not biased since this was not performed by individuals blind to the experiment who were likely, not unfamiliar with the different gestation stages in marmoset development?

Thank you for asking this question. Such a bias is unlikely because:

The scoring process was done blind to gestational age (the videos were not tagged in any way with gestational age).An accurate gestational age of the fetus cannot be determined by eye alone. The ultrasound probe placement (angle, distance from the fetus etc.) can change the apparent of size of the fetus. To determine gestational age, we needed to measure the biparietal distance of the fetal skull.The videos were scored long after (several months) they were acquired. The analysis of these scores were done long after (several months) after the scoring. Indeed, we were surprised to discover the head/mouth movement link of fetal marmosets.The videos were scored by 3 individuals, with every alternate day being scored by a different person. Two of the 3 scorers were not involved in formulating the hypothesis or doing the analysis.

We have now included these details in our Methods section.

2. The text is insufficiently clear in defining how state transitions are defined. Please clarify whether Shannon entropy and the KL divergence are just used for quantifying the relative proportion of each state rather than for quantifying the state changes per se. Further details of how the information theory analysis is affected by state probability are needed.

We estimated the first order Markov chain to generate the state transitions (Figure 2B).

We used Shannon Entropy (Figure 2C) and Kullback-Leibler Divergence (Figure 2D) to quantify the relative proportion of each state in each session and compare over the course of gestation.

To clarify these points, we made the following modifications in the manuscript:

In the Results section we have now written: “We quantified these state changes across gestational days in two ways. First, Shannon entropy was used to measure behavioral variability within each session (Figure 2C).”

We have also included more details in the methods section under subsection “Information theory analysis – testing for increase in ‘order’ through gestation”.

3. The criteria for defining the timing of head and orofacial movements were felt to be too rigid, so some exploration of this was felt to be desirable.

To explore the possibility that our results are dependent on the state criteria we set (including the criteria for synchronicity — state 5), we analyzed the data using durations instead of counting movement occurrences. The results of this exploration are in the Author response image 1, 2 and 3. They show that the patterns we presented using movement occurrences (defined more clearly in our response #2 above) is preserved using duration as well. Thus, our results were not dependent on the state criteria. In the revised manuscript, we chose to define states in the original manner, using movement occurrences, since this allowed us to do the information theory analysis.

**Author response image 1. sa2fig1:** Developmental profile of orofacial movements using duration as the measure. Y axis: total duration of orofacial movements per session / total duration fetal face is visible per session; X axis: gestational age.

**Author response image 2. sa2fig2:** Developmental profile of head movements using duration as the measure. Y axis: total duration of head movements per session / total duration fetal face is visible per session; X axis: gestational age.

**Author response image 3. sa2fig3:** Overlap decline. Y axis: total overlap duration of orofacial and head movements per session / total duration of orofacial and head movements per session; X axis: gestational age.

4. The KL divergence is quantified with respect to the first post-natal day. This may make this measure quite noisy. The effects of averaging over the first few days should be considered.

Thank you for bringing this to our notice. As the review suggested, taking the average made our results more robust. We have modified the manuscript accordingly.

The text in the Results section now reads:

“Second, we performed a Kullback-Leibler divergence test to quantify the behavioral change through fetal life (Figure 2D). The average state distribution of E93-99 (n=6) was compared with the state distributions of all the imaging days. The best polynomial-fit order for the divergence estimates was one. The resulting linear fit with a positive slope (p<0.001) indicates that with increasing gestational age, fetal behavior – with respect to orofacial and head movements – becomes increasingly different from the first imaging day of orofacial movement.”

5. Please consider whether it is possible that an increase in lick-like movements over the course of development could have been missed because of a ceiling effect.

Yes, it is possible that an increase in lick-like movements over the course of development could have been missed because of a ceiling effect. It could be that the lick-like movements develop at an earlier time in gestation. Based on our observation, what we can conclude is that in the gestation period under observation (~E93 to birth), there is no significant increase in lick-like movements. Since our focus was on the development of contact calls, we did not probe further. We added this note to the Results.

6. While we appreciate that you have attempted to show that orofacial movements are specifically related to vocal production, it was felt that this point needs to be more nuanced. Specifically, orofacial movements may be also important for sucking or breathing and are therefore part of the overall development of bodily movements. Furthermore, previous studies have shown that marmoset monkeys produce contact calls around every 10 s (~0.1 Hz), which are correlated with heart rate and "Mayer wave", an oscillation of the autonomic nervous system. Please address the concern that the link between orofacial movements and the production of contact calls could be explained by autonomic responses, such as breathing.

In the current manuscript, we focused on a single type of orofacial movement—those related to contact call/crying-like vocal production. We occasionally certainly observed other types of mouth movement but they were too subtle to classify and/or quantify. That is, while we’re confident movements like sucking are present at the fetal stage in marmosets (after all, suckling is a requirement right after birth), we could not discern them. That’s what makes the contact call movement special and our focus—no other orofacial movement is similar or more distinct. We make no claims about other orofacial movements and their development (beyond licking and twitter calls).

Based on our previous work (Zhang and Ghazanfar, 2016), we know that the start of infant vocal sequences are linked to a 0.1Hz autonomic rhythm and that the temporal structure of those sequences are correlated with the respiratory cycle. Thus, one possibility is that the timing of fetal bodily and/or orofacial movements are also tied to such cycles, or at least to the autonomic rhythm (in humans at least, fetal “practice” breathing movements are intermittent). Early on in our study, we attempted to investigate this link but we could not simultaneously monitor heart rate (in mothers or fetuses) while also monitoring orofacial movements.

Nevertheless, we now consider this possible link between fetal orofacial movements to autonomic rhythms in the Discussion.

7. If possible, please comment on whether there is any evidence for the production of "cry" calls, similar to those of human babies, during the prenatal stage.

Like many (if not all) mammals, marmosets and human in early infancy produce cries or “contact” vocalizations when separated from parents. So, in that respect, they are similar. In addition, however, our previous study (Zhang and Ghazanfar, 2016) showed that the early infant vocal production is both rhythmical and linked to arousal, as it is in humans (Zeskind and Lester, 1978). Finally, the orofacial movements are strikingly similar. For these reasons, it seemed reasonable to claim that marmoset and humans produce similar “cry” calls.

8. The apparent inconsistency with your previous work over whether marmosets produce contact calls immediately after birth needs to be addressed (see reviewer 3 comments).

Thank you for this comment. However, there is no inconsistency with our previous work. In our first study of vocal development in marmosets (Takahashi et al., 2015), we showed that what others were classifying as separate “cry” calls were really just immature versions of the contact call. Early in postnatal life, marmosets produce mostly immature contact calls and then gradually begin to exclusively produce adult-like contact calls (Takahashi et al., 2015; Zhang and Ghazanfar, 2016; Zhang and Ghazanfar, 2018; Zhang et al., 2019). While the immature and mature versions sound different (one is noisy, the other is very tonal), the difference in sound is related to lung and laryngeal development, not orofacial movements. The orofacial movements associated with immature and mature contact calls are indistinguishable. Thus, in the current study, we use the term “contact call” referring to both immature and mature contact calls. We clarify this in the Introduction of the revised manuscript.

9. The statistical results should be shown for the distributions of each monkey, to demonstrate how reliable these correlations are for each individual.

We have now included the statistics for each pregnancy in the supplementary section:

The sample size for each pregnancy is small (limited naturally by the gestation period of the marmoset). Though we did see significance at the population level--even after controlling for pregnancies--we did not have enough statistical power for individual pregnancies. Nevertheless, we report those results here. Importantly, the sign of the effect (β) is consistent for each pregnancy.

Matching temporal profiles of fetal orofacial movements to that of week 1 infant contact calls (related to Figure 4C)

Pregnancy 1 β±SE = -0.01±0.01, t = -1.28; F(1,10)=1.64; p=.23

Pregnancy 2 β±SE = -0.003±0.002, t = -1.38; F(1,15)=1.90; p=.19

Pregnancy 3 β±SE = -0.008±0.007, t = -1.11; F(1,14)=1.23; p=.29

Pregnancy 4 β±SE = -0.003±0.005, t = -0.53; F(1,14)=0.28; p=.60

Matching fetal orofacial movements to infant call profile (related to Figure 4I)

Pregnancy 1 β±SE = 0.03±0.13, t = 0.25; F(1,13)=0.06; p=.80

Pregnancy 2 β±SE = 0.27±0.08, t = 3.22; F(1,16)=10.38; p=.006

Pregnancy 3 β±SE = 0.18±0.08, t = 2.21; F(1,16)=4.89; p=.04

Pregnancy 4 β±SE = 0.08±0.06, t = 1.17; F(1,15)=1.36; p=.26